# Label-free automated three-dimensional imaging of whole organs by microtomy-assisted photoacoustic microscopy

Terence T. W. Wong [1,2], Ruiying Zhang[2], Chi Zhang[2], Hsun-Chia Hsu[1,2], Konstantin I. Maslov[1], Lidai Wang[2,5], Junhui Shi[1], Ruimin Chen [3], K. Kirk Shung[3], Qifa Zhou[3,4] & Lihong V. Wang[1]

Three-dimensional (3D) optical imaging of whole biological organs with microscopic resolution has remained a challenge. Most versions of such imaging techniques require special preparation of the tissue specimen. Here we demonstrate microtomy-assisted photoacoustic microscopy (mPAM) of mouse brains and other organs, which automatically acquires serial distortion-free and registration-free images with endogenous absorption contrasts. Without tissue staining or clearing, mPAM generates micrometer-resolution 3D images of paraffin- or agarose-embedded whole organs with high fidelity, achieved by label-free simultaneous sensing of DNA/RNA, hemoglobins, and lipids. mPAM provides histology-like imaging of cell nuclei, blood vessels, axons, and other anatomical structures, enabling the application of histopathological interpretation at the organelle level to analyze a whole organ. Its deep tissue imaging capability leads to less sectioning, resulting in negligible sectioning artifact. mPAM offers a new way to better understand complex biological organs.

[1] Caltech Optical Imaging Laboratory, Andrew and Peggy Cherng Department of Medical Engineering, Department of Electrical Engineering, California Institute of Technology, Pasadena, CA 91125, USA. [2] Optical Imaging Laboratory, Department of Biomedical Engineering, Washington University in St. Louis, St. Louis, MO 63130, USA. [3] NIH Resource Center for Medical Ultrasonic Transducer Technology, Department of Biomedical Engineering, University of Southern California, Los Angeles, CA 90089, USA. [4] Roski Eye Institute, Department of Ophthalmology and Biomedical Engineering, University of Southern California, Los Angeles, CA 90089, USA. [5]Present address: Department of Mechanical and Biomedical Engineering, City University of Hong Kong, Hong Kong, China. Terence T.W. Wong and Ruiying Zhang contributed equally to the work. Correspondence and requests for materials should be addressed to L.V.W. (email: LVW@Caltech.edu)

In biomedical imaging, all optical techniques face a fundamental trade-off between spatial resolution and tissue penetration; hence, obtaining an organelle-level resolution image of a whole organ has remained a challenging and yet appealing scientific pursuit. Over the past decade, optical microscopy assisted by mechanical sectioning or chemical clearing of tissue has been demonstrated as a powerful technique to overcome this dilemma, one of particular use in imaging the neural network[1–6]. Thanks to recent advances in computing power, the acquired data, typically terabytes in size, can be automatically processed to visualize the three-dimensional (3D) structure in a whole brain. However, this type of techniques needs lengthy special preparation of the tissue specimen, which hinders broad application in life sciences. For example, diffusion staining of a whole brain[3] is extremely slow due to the scant extracellular space in the central nervous system. Similarly, electrophoretic removal of lipids in the brain[6], resulting in a transparent brain for easy staining and imaging, causes an uncertain loss of biological information. Therefore, finding an imaging method applicable to minimally processed tissue, ideally fresh tissue, can provide new insights into complex biological systems and make whole-organ microscopy a universal laboratory technique.

Among all 3D imaging techniques, histology is an attractive way to analyze specimens because histopathological interpretation can be readily applied from organelle to organ levels. However, ordinary wide-field optical microscopy cannot provide optical sectioning, resulting in blurry images when a thick tissue is imaged. To get high resolution and high contrast histologic images, thick tissue always requires mechanical sectioning before imaging. Therefore, 3D histology can be obtained only by registering images of all thin slices, each subjected to a different level of inaccuracy, despite considerable rectification efforts in the field[7–12]. In addition, sectioning before imaging can also cause undesired tissue ruptures in each thin slice, further reducing the quality of registered images. So far, sharply imaging the presented surface of tissue before sectioning it remains a challenging and yet appealing goal in histology.

Here, we propose a new label-free 3D imaging technique, named microtomy-assisted photoacoustic microscopy (mPAM), for potentially imaging all biomolecules with 100% endogenous natural staining in whole organs with less sectioning and high fidelity. Photoacoustic (PA) microscopy (PAM)[13,14] is a fast developing label-free imaging method. While in an unstained piece of tissue, most endogenous biomolecules do not fluoresce; however, all of them absorb photons at some wavelengths. Most absorbed light energy will be converted into heat, which results in an acoustic pressure rise propagating as ultrasound—the signal source for PAM. Label-free PAM has been demonstrated in such broad biomedical applications as imaging DNA/RNA[15], cytochromes[16], hemoglobins[17], melanin[18], and lipids[19] at an optical wavelength ranging from ultra-violet (UV) to near-infrared. This list of imaging targets is expanding with the ongoing exploration of endogenous absorption, and label-free imaging of all biomolecules remains possible. Moreover, PAM in reflection mode is applicable to large tissue volumes and does not require preparation of thin tissue sections[1]. Combined with a microtome for serial removal of previously imaged tissue sections, PAM performs well as a tool for imaging biomolecules of interest in an unstained organ at subcellular resolution. Furthermore, PAM's label-free nature enables it to image differently embedded organs for different applications, e.g., paraffin and agarose are the most common embedding materials used in conventional histology and neuroscience[20], respectively. In this paper, we demonstrate the first label-free mPAM, using UV light for histology-like

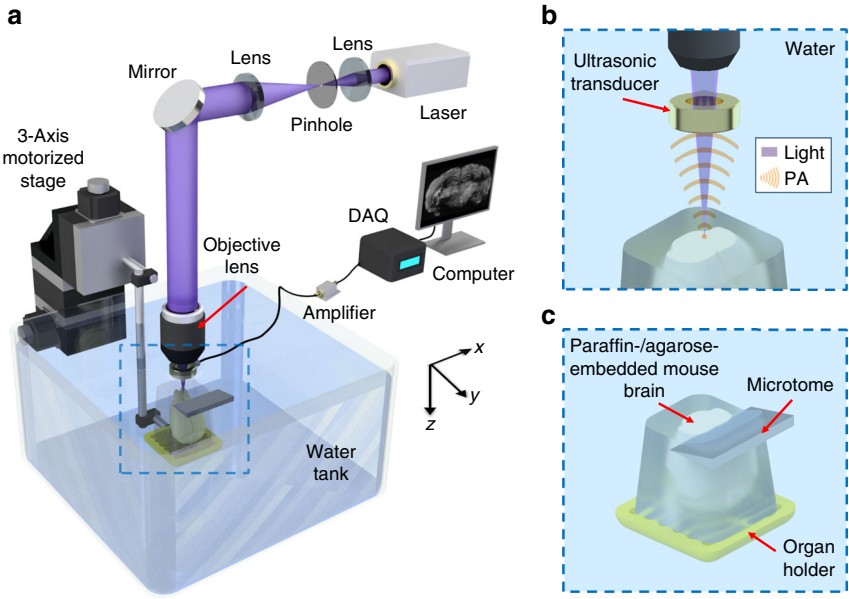

**Fig. 1** Schematic of the mPAM system for whole-organ imaging and sectioning. **a** The UV laser beam is first spatially filtered and expanded by a pair of lenses and a pinhole. The beam is then focused through an objective lens and passed through a ring-shaped ultrasonic transducer onto the surface of the paraffin-/agarose-embedded organ (e.g., a mouse brain), which is placed inside a water tank on top of a sample holder. Some generated acoustic waves propagate backward and reach the ring-shaped focused ultrasonic transducer. The received acoustic pressure is transduced into an electric signal, which is then amplified and recorded by a data acquisition (DAQ) card. During data acquisition, a maximum amplitude projection image from the measured B-scan data is displayed on a computer screen in ~1 s. By raster scanning the sample holder, a maximum amplitude projection image of the exposed tissue surface is also acquired. The imaged surface is then sectioned by a microtome, and a new surface is imaged automatically. This process continues until the sectioned layers reach the preset depth. **b** Close-up of the blue dashed region in **a** during imaging. The UV light passes through the ring-shaped focused ultrasonic transducer, inducing acoustic waves which are partially back-propagated and received by the same ultrasonic transducer. **c** Close-up of the blue dashed region in **a** during sectioning. The imaged surface's (cross-section) is being cut by the microtome

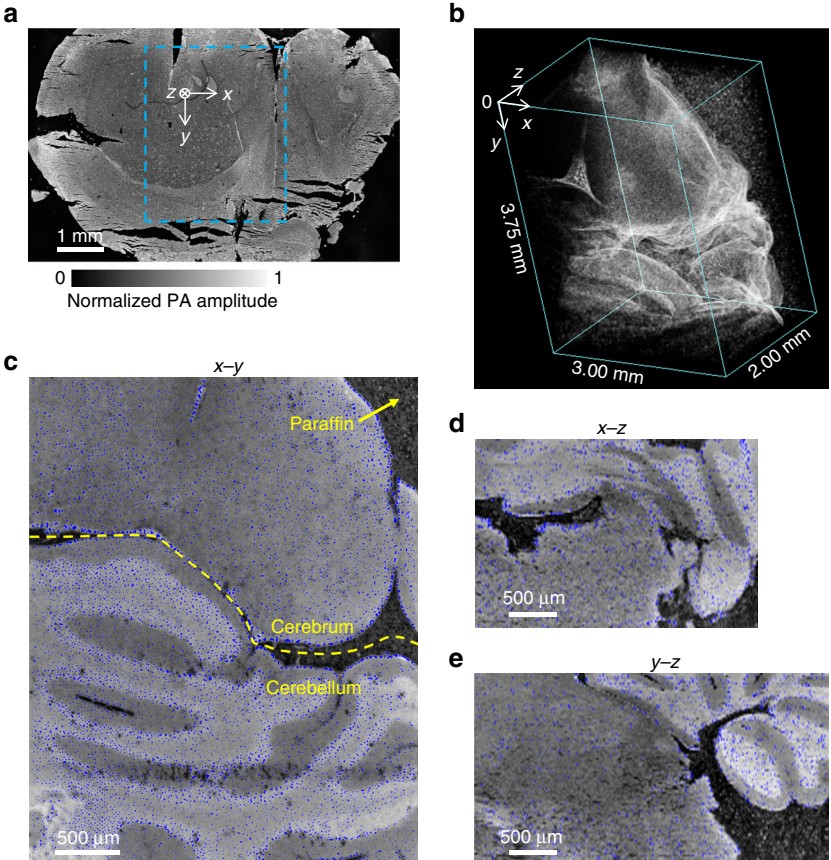

**Fig. 2** 3D label-free mPAM image of an unstained mouse brain embedded in a paraffin block. **a** A section of the entire mouse brain image (coronal view). **b** A 3D view of the imaged brain block corresponding to the marked region in **a**. **c** x–y image at z = 0.16 mm (coronal view), with the cell nuclei marked in blue. The yellow dashed line outlines the boundary between the cerebrum and the cerebellum. **d** x–z image at y = 2.31 mm (transverse view). **e** y–z image at x = 0.63 mm (sagittal view)

imaging without staining[21], in whole organs (e.g., mouse brains), most of them formalin-fixed and paraffin- or agarose-embedded for minimal morphological deformation. A fresh mouse kidney embedded in agarose is also imaged. In addition, mPAM with dual wavelength illuminations is also employed to image a mouse brain slice, demonstrating the potential for label-free imaging of multiple biomolecules. With visible light illumination, mPAM shows its deep tissue imaging capability, which enables less slicing and hence reduces sectioning artifacts.

## Results

**mPAM system for whole-organ imaging and sectioning.** In mPAM (Fig. 1a), an organ (e.g., a mouse brain) or a tissue block, either formalin-fixed or fresh, and paraffin or agarose embedded, is mounted on an organ holder immersed in water. The tissue is automatically imaged under a computer control. A laser generates pulses at 266 nm wavelength (and 420 nm for dual wavelength illumination) to predominantly excite DNA/RNA in the tissue, and the generated PA waves are detected by a ring-shaped focused ultrasonic transducer (Fig. 1b). The 3-axis motorized stage controls both the scanning for imaging and the tissue sectioning by the microtome. The mPAM system records and displays the cross-sectional images (e.g., coronal sections of a mouse brain) in real time during data acquisition. The exposed top tissue surface is imaged, then a thin layer is shaved off (Fig. 1c), and the new surface is imaged. This sequence is repeated to obtain a 3D image. The mPAM system currently provides a lateral resolution of 0.91 μm (Supplementary Fig. 1), more than sufficient to image individual cell nuclei without labeling. Moreover, our mPAM

system can handle organs of various sizes because it is implemented in reflection mode.

**Imaging a formalin-fixed paraffin-embedded mouse brain.** First, we validated the mPAM system by imaging a formalin-fixed thin paraffin section of a mouse brain (Supplementary Fig. 2). The unstained paraffin section, fixed on a quartz slide that is UV transparent, was imaged by mPAM (Supplementary Fig. 2a) and then stained with hematoxylin and eosin (H&E) (Supplementary Fig. 2b) for comparison with conventional histology. The corresponding close-up images are shown in Supplementary Fig. 2c, d, respectively. The cell nuclei in the mPAM image were enhanced by Hessian filtering (Supplementary Fig. 2a, c) and are highlighted in blue. The step-by-step cell nuclear extraction results of the Hessian filtering are shown in Supplementary Fig. 3. The gray matter and white matter can be differentiated in the mPAM image because the former has a higher density of nuclei than the latter. The nuclei in the mPAM image match well with those in the H&E image. Using the H&E image as the gold standard, in identifying nuclei, mPAM has a sensitivity of 93.2%, a specificity of 99.8%, and a positive predictive value (PPV) of 96.7%. This experiment shows that mPAM can pinpoint cell nuclei sensitively and specifically in a paraffin-embedded organ section.

Next, we used mPAM to image a formalin-fixed and paraffin-embedded mouse brain block. Similar to imaging a thin section, a Hessian filter was employed for cell nuclear extraction (Supplementary Fig. 4a). Compared with imaging a thin section, imaging an unstained paraffin block resulted in a stronger background, and thereby a lower image contrast for nuclei. The stronger

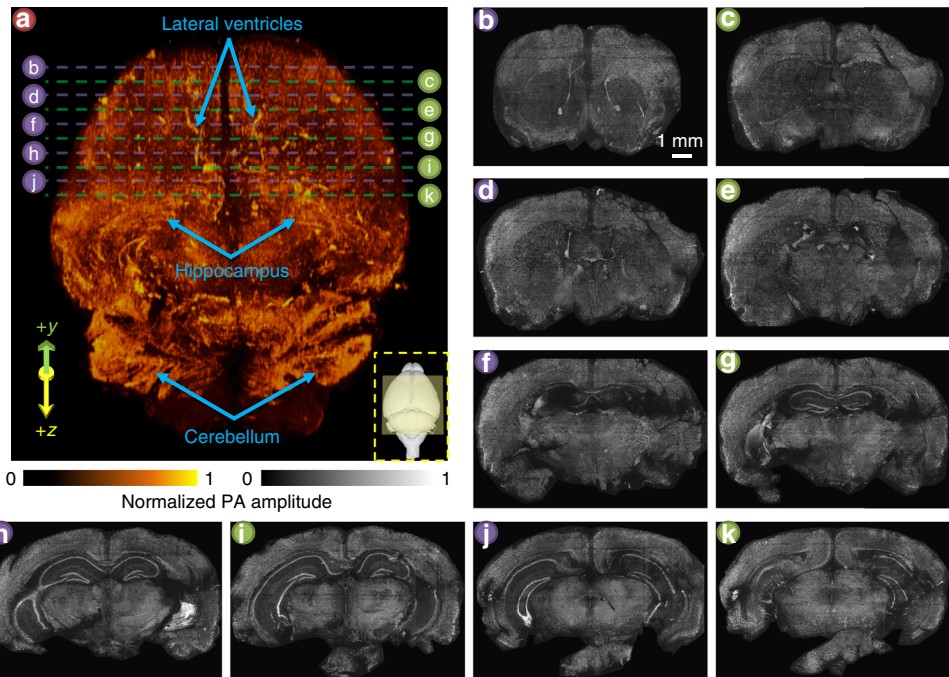

**Fig. 3** 3D label-free mPAM image of an unstained mouse brain embedded in an agarose block. **a** An mPAM volumetric image of a mouse brain with a sectioning step size of 200 μm. The bottom right yellow dashed inset shows the corresponding projection and its imaging area (yellow shaded region) of a 3D mouse brain model (reprinted from Elsevier, Vol. 53, Johnson, G.A., Badea, A., Brandenburg, J., Cofer, G., Fubara, B., Liu, S. & Nissanov, J., Waxholm space: an image-based reference for coordinating mouse brain research, 365–372, copyright (2011), with permission from Elsevier)[33]. Three features, namely the lateral ventricles, hippocampus, and cerebellum, are labeled on the 3D mPAM image. **b–k** The relative positions of ten coronal views are labeled in **a**, and they are shown individually. The separation between each section is 500 μm

background from paraffin caused false positive cell nuclear identifications, which could be eliminated by calculating the nuclear mask of the mPAM image (Supplementary Fig. 4b, c). Due to UV-light attenuation, the nuclear signal was expected to decrease exponentially with depth. To estimate the mPAM imaging depth in the block (Supplementary Fig. 5a), which is related to the selection of sectioning thickness in 3D mPAM, the mouse brain block was sectioned at the surface by a standard microtome for quantitative analysis. We obtained a series of H&E images of these sections, each 7 μm thick (Supplementary Fig. 5b–d). Due to the deformation caused by sectioning, the nuclei in the H&E section images cannot be matched exactly with those in the mPAM block image (as we did in Supplementary Figs. 2 and 3). However, the distributions of nuclei in the mPAM and H&E images are strongly correlated. To quantify this correlation, the nuclear count and nuclear density were calculated for these images (Supplementary Fig. 5e–h). The ratio of the nuclear count in the H&E images within a given depth range to that in the mPAM image was calculated to be closest to unity for a depth range of 21 μm (Supplementary Fig. 5i). The correlation coefficient was calculated between the nuclear density map of the H&E images within a given depth range and that of the mPAM image, yielding a maximum of 0.78 over a depth range of 14 μm (Supplementary Fig. 5j). In fact, the sensitivity of mPAM to nuclei decreases gradually with depth, depending on both the light attenuation with depth and the absorption coefficients of different nuclei, but this phenomenon is difficult to model accurately and so is not taken into account. Given the values of the nuclear count ratio and the correlation coefficient, we estimated that mPAM imaged 14–21 μm deep in the block.

Next, we demonstrated the full capacity of mPAM for 3D high-resolution imaging (Fig. 2). An unstained mouse brain block (as used previously) was imaged on the surface (Fig. 2a) and sectioned repeatedly and re-imaged at 20 μm thickness by

mPAM. The imaged volume of 3.0 mm by 3.8 mm by 2.0 mm (Fig. 2b) took ~70 h for data acquisition. The volume covered both the cerebrum and the cerebellum (Fig. 2c). Representative $x$–$y$, $x$–$z$ and $y$–$z$ images are shown in Fig. 2c–e, respectively. Cell nuclei are highlighted in blue. As the images were acquired at the block surface before sectioning, mPAM did not present artifacts of deformed or discontinuous structures, which are common problems in histology. In Supplementary Fig. 2b, for example, deformation is especially evident at the bottom. The serial two-dimensional (2D) images were combined into a 3D image without the need for image registration. Supplementary Movie 1 shows the 3D mPAM image of the mouse brain, revealing individual cell nuclei clearly in coronal, transverse, and sagittal views, and stepping through different positions and view angles.

**Imaging formalin-fixed agarose-embedded mouse brains.** We used agarose-embedded organs to further demonstrate applying mPAM to different embedding materials. The advantages of using agarose-embedding are two-fold: (i) Unlike paraffin, agarose is highly UV transparent and does not infiltrate into tissue[22], which improves the imaging contrast of mPAM, and (ii) agarose can be used for embedding fresh tissue[23], which enables mPAM to be readily used in life science studies. The only drawback is that agarose is a softer embedding material than paraffin, which limits obtaining thin sections. Nevertheless, a section thickness as small as 50–100 μm can be achieved with the integration of a vibratome[24], which is enough for numerous applications.

As an initial validation, a formalin-fixed paraffin-embedded mouse brain slice was imaged by mPAM and used as a reference (Supplementary Fig. 6a). The slice was then deparaffinized, embedded in agarose, and re-imaged by mPAM (Supplementary Fig. 6b). For comparison, an adjacent brain section was H&E stained and imaged by a conventional wide-field microscope

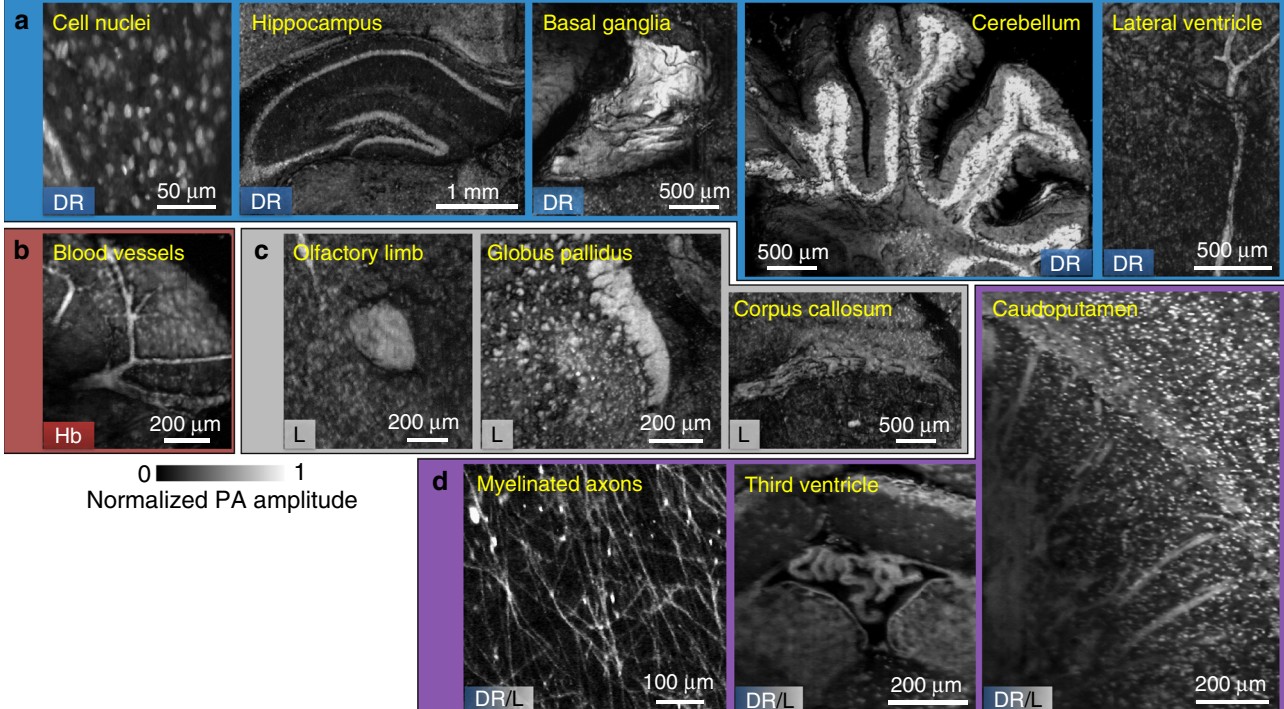

**Fig. 4** Image gallery of features extracted from label-free mPAM images of four unstained mouse brains embedded in agarose blocks. All features are shown in coronal view. Collections of images showing the biomolecules that provide absorption contrast due to **a** DNA/RNA (DR), **b** hemoglobins (Hb), and **c** lipids (L). **d** Images of myelinated axons, third ventricle, and caudoputamen due to both DNA/RNA and lipids contrasts

(Supplementary Fig. 6c). The corresponding close-up images are shown in Supplementary Fig. 6d–f, respectively. The close-up images clearly show that a deparaffinized and agarose-embedded brain slice can reveal individual cell nuclei without any contrast enhancement algorithm, which was also validated with the H&E close-up image. Paraffin is a UV absorbing material that infiltrates tissue. Thus, when it was washed out (i.e., deparaffinized), the background signal was reduced, and the contrast of all the original tissue structures was boosted (Supplementary Fig. 6d, e). This experiment shows that with agarose embedding, mPAM can pinpoint cell nuclei without any contrast enhancement algorithm, further improving the accuracy of cell nuclear identification.

As a proof of concept with a microtome, we imaged an entire formalin-fixed agarose-embedded mouse brain, with a 200 μm section thickness. The imaged volume of 9.5 mm by 7.5 mm by 11.0 mm (Fig. 3a) took ~15 days for data acquisition. The lateral ventricles, hippocampus, and cerebellum can be observed in the 3D image, which was obtained by stacking 56 coronal sections of the mouse brain (Fig. 3a). Each individual coronal section with its corresponding position in the mouse brain is shown in Supplementary Movie 2. Close-up images of a row of representative coronal sections are shown in Supplementary Movie 3 to illustrate the high-imaging resolution of mPAM. To show high-quality coronal sections without the integration of a vibratome, which is the ideal sectioning tool for agarose-embedded organs[22], we used a larger sectioning thickness of 500 μm (Fig. 3b–k). The sections' corresponding positions are labeled in the 3D mouse brain mPAM image (Fig. 3a).

To further show the strength of label-free mPAM that many biologically important features of the mouse brain could be imaged, two more formalin-fixed agarose-embedded mouse brains were imaged with different sectioning thicknesses, 300 and 400 μm. Together with the aforementioned 200 and 500 μm sectioning thicknesses, we covered biological features that can be

found in every 200, 300, 400, and 500 μm. Figure 4 shows a collection of images of features extracted from all four agarose-embedded mouse brains. These features clearly reveal the unique capability of label-free mPAM, which allows imaging of different biomolecules that otherwise would require different labeling/dyes for simultaneous visualization. For instance, the leftmost mPAM images in Fig. 4a–c show cell nuclei, blood vessels, and the olfactory limb by imaging DNA/RNA, hemoglobins, and lipids with UV light illumination alone. Moreover, features such as myelinated axons and third ventricle can be imaged without staining due to their rich DNA/RNA and lipid contrasts (Fig. 4d). By illustrating all the features with a single imaging modality and minimal tissue processing, mPAM enables understanding and exploring the structural or connection changes of different biological structures under different conditions, such as different diseases and stages, with high fidelity. To increase the imaging specificity, multi-wavelength illumination can be used to decouple signal contributions from different biomolecules[25], with each illumination wavelength falling into a strong absorption band of a biomolecule. To show that mPAM can image more endogenous biomolecules, we used dual-wavelength illumination (266 and 420 nm) to image an agarose-embedded brain slice (Supplementary Fig. 7a, b). With 266 nm laser illumination, Supplementary Fig. 7a shows mostly DNA/RNA and lipid contrasts, whereas with 420 nm laser illumination, Supplementary Fig. 7b shows mostly cytochrome contrast. The overlay image (Supplementary Fig. 7c) is displayed in two-channel pseudo colors, which represent the optical absorption color contrasts of the biomolecules at the two wavelengths and illustrates that more biomolecules are imaged by dual-wavelength mPAM than by single-wavelength mPAM. To show that deep tissue imaging can also be achieved with mPAM, we used 420 nm light to illuminate a 1 mm thick mouse brain slice. A representative xz projected image of the mouse brain is provided in Supplementary Fig. 8, which shows cytochrome contrast based structures ~800 μm

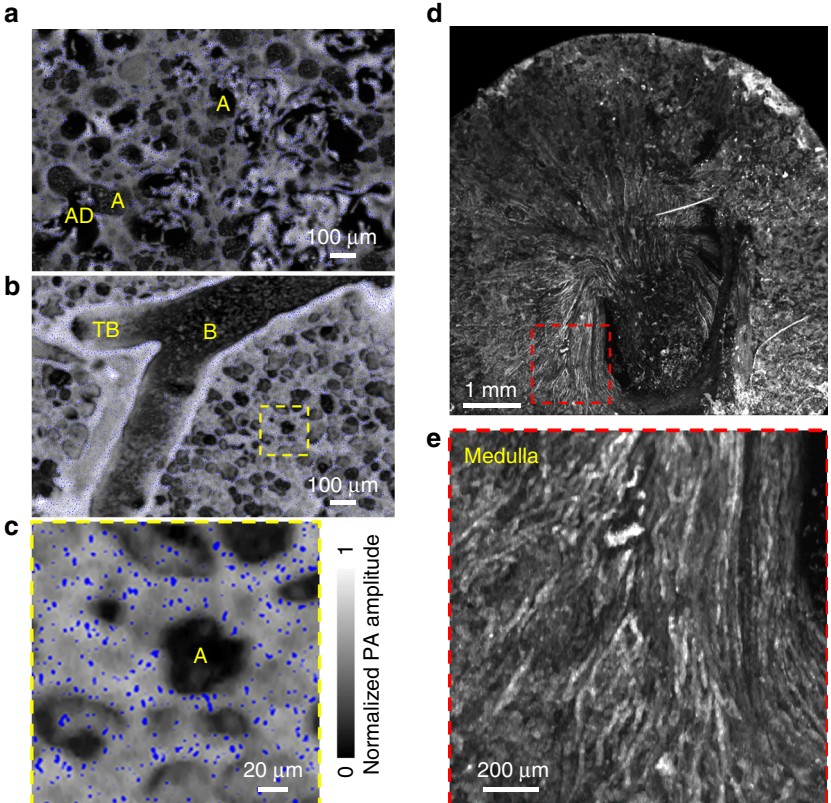

**Fig. 5** Label-free mPAM images of an unstained formalin-fixed mouse lung embedded in a paraffin block and an unstained fresh mouse kidney embedded in an agarose block. **a**, **b** Two *x*–*y* sections of the mouse lung. **c** Close-up image of the yellow dashed region in **b**. A, alveoli; AD, alveolar duct; B, bronchiole; TB, terminal bronchiole. **d** A *x*–*y* section of the mouse kidney. **e** Close-up image of the red dashed region in **d**

deep. Therefore, where cytochrome is the target of interest, we can further reduce the number of slices, resulting in even fewer sectioning artifacts.

**Imaging a formalin-fixed paraffin-embedded mouse lung and a fresh agarose-embedded mouse kidney.** Finally, to demonstrate that label-free mPAM can image different organs and even fresh tissue, we imaged a mouse lung and kidney. The mouse lung was formalin fixed and embedded in paraffin, and the fresh mouse kidney was embedded in agarose. For the mouse lung, two mPAM images of sections show different recognizable features (Fig. 5a, b). A close-up mPAM image of the paraffin-embedded mouse lung (Fig. 5c) shows that individual cell nuclei can be extracted (in blue). Similarly, to show the high-resolution 3D imaging capability of mPAM, we imaged 61 sections with a 20 μm section thickness. The imaged volume was 1.5 mm by 1.0 mm by 1.2 mm. Cell nuclei are shown clearly in the coronal view in the video (Supplementary Movie 4). For the fresh mouse kidney, an entire section was imaged by mPAM (Fig. 5d). A close-up mPAM image (Fig. 5e) clearly shows the typical fiber-like structures in the medulla region of the mouse kidney. These results indicate that mPAM can provide histology-like imaging of organs that are either formalin-fixed or fresh, and paraffin or agarose embedded.

## Discussion

mPAM offers a new way to analyze disease-induced structural changes or the system function of a whole organ. By imaging cell nuclei and blood vessels, mPAM can also serve as a minimal-artifact substitute for histology. mPAM facilitates rapid 3D imaging of large tissue specimens. It can be readily applied to most

standard paraffin blocks used in histology, e.g., paraffin blocks of an entire mouse brain or lung (Figs. 2, 5; Supplementary Movies 1, 4). Such large volume registration-free histologic 3D imaging is impossible with any of the current choices for whole-organ microscopy. Moreover, mPAM can also be applied to fixed or fresh agarose-embedded tissue imaging (Figs. 3–5; Supplementary Movies 2, 3), and so should find broad applications in basic life science studies.

mPAM is currently in the early stages of development, and significant technical improvements will be realized in the future. First, the laser repetition rate of only 10 kHz made it very slow to image a whole organ (e.g., a mouse brain). In our experiment, it took ~70 h to image about one-twentieth of the volume of a mouse brain, and ~15 days to image an entire mouse brain which was not densely sectioned. With the combination of a fast laser[26] and a fast scanning mechanism[17,27] in the future, the imaging speed is expected to be increased by two orders of magnitude, achieving subcellular imaging of a whole densely sectioned mouse brain within one day. In addition, by implementing multiple channels using a microlens array[28,29], the acquisition can be accelerated by additional orders of magnitude. Second, by incorporating the capability to employ more and different wavelengths in the future, mPAM can potentially probe many more endogenous biomolecules and specific cells, such as only neurons in brains, at their absorption peak wavelengths. Third, with the integration of a vibratome in the mPAM system, it is possible to achieve high-quality and densely sectioned images, even for agarose-embedded fresh tissue. With these further developments, mPAM may become a universal laboratory technique for whole-organ microscopy, with diverse applications in life sciences.

## Methods

**Organ preparation**. The organs were extracted from Swiss Webster mice (Hsd: ND4, Harlan Laboratories). The brain, one lung, and the kidney were harvested immediately after each mouse was sacrificed. The brain and lung were fixed in 10% neutral-buffered formalin at room temperature for 5 days. Afterwards, the brain and the lung were embedded in paraffin as block specimens, following standard histology procedure, and then sectioned by a microtome into thin slices as required. Four more brains were embedded in 4% agarose as block specimens, and then sectioned by a microtome into thin slices with different sectioning thicknesses. The fresh kidney was sectioned by hand to a ~1 mm thick slice, then embedded in 4% agarose as a block specimen. All experimental animal procedures were carried out in conformity with a laboratory animal protocol approved by the Animal Studies Committee of Washington University in St. Louis.

**mPAM system**. The user interface of the mPAM system is programmed in Lab-VIEW. After acquiring all the inputs from the user, the computer transfers all the parameters to a central controller (sbRIO-9623, National Instruments) which integrates a real-time processor (400 MHz) and a reconfigurable field-programmable gate array. The controller triggers an Nd:YLF Q-switched UV laser (QL266-010-O, CrystaLaser) to generate laser pulses with a 266 nm wavelength, 7 ns pulse width, ~5 nJ pulse energy on the imaging targets, and 10 kHz pulse repetition rate, or an OPO laser (NT242-SH, Altos Photonics) to generate laser pulses with a 420 nm wavelength, 5 ns pulse width, ~200 nJ pulse energy on the imaging targets, and 1 kHz pulse repetition rate. The laser beam is focused onto the organ immersed in water by a custom-made water-immersion UV objective lens (consisting of an aspheric lens, a concave lens, and a convex lens (NT49-696, NT48-674, NT46-313, Edmund Optics)); Supplementary Fig. 1a) with a numerical aperture (NA) of 0.16. The excited photoacoustic waves from the organ are detected by a custom-made ring-shaped ultrasonic transducer (42 MHz center frequency, 76% −6 dB bandwidth), which has a central hole for light delivery. The signals are then amplified, digitized by a data acquisition card (installed on the computer and triggered by the controller; ATS9350, Alazar Technologies), recorded on the computer hard disk, and displayed in real time on the computer screen. The controller also triggers the scanning stages (x and z stages: PLS-85, PI miCos, GmbH; y stage: LS-180, PI miCos, GmbH), in synchronization with the laser, for point-by-point imaging of the organ surface. By calculating the amplitude of each A-line photoacoustic signal, we obtain a 2D image of the specific optical absorption $(J/m^3)$ of the organ. After each surface image is acquired, the organ, controlled by the scanning stages, is automatically sectioned by a microtome blade mounted inside the water tank. The sliced-off paraffin-embedded thin sections of the organ float to the water surface and are confined within a specific area. The sliced-off agarose-embedded thin sections sink to the bottom of the water tank. The imaging and sectioning process is repeated as required. Later, the serial 2D images are processed for 3D visualization. Note that, to avoid laser overload, we paused the laser for 30 min between consecutive raster scans. We did not observe any power drop of the laser throughout acquisition of all the volumetric images. With our current design and setting, the imaging speed is limited by the laser repetition rate to $10^4$ pixels/s. The lateral scanning step size is 0.625 μm. At $10^4$ pixels/s, the required scanning speed of a motorized stage is ~6.25 mm/s. The motorized stage that we used for the fast scanning axis (PLS-85) can scan at up to 50 mm/s. Therefore, the motorized stage is not the limiting factor on the imaging speed.

**Lateral and axial resolution measurements**. The mPAM system's lateral resolution is determined by the optical focusing, because in the focal plane only those biomolecules inside the optical focus are excited[30]. According to Zemax simulation, the UV objective lens can provide a diffraction-limited resolution as fine as 0.34 μm at 0.4 NA. But in practice the optical NA is limited to 0.16 by the size of the central hole of the ring-shaped ultrasonic transducer. Accordingly, the lateral resolution is 0.91 μm, as validated by experiments (Supplementary Fig. 1b). The axial resolution of a linear photoacoustic system is determined by the bandwidth of the ultrasonic transducer[31] (estimated as 25 μm for mPAM). However, here the strong UV absorption from embedded organs is likely to be the limiting factor for axial resolution, because the estimated imaging depth is ~20 μm (Supplementary Fig. 5). Thus the axial resolution of mPAM is ~20 μm.

**Image processing**. We designed a Hessian filter to mathematically extract the cell nuclei from the 2D mPAM images (Supplementary Fig. 3). For a 2D image function f(x, y), a Hessian matrix was constructed for each pixel[32]:

$$H = \begin{bmatrix} \frac{\partial^2 f}{\partial x^2} & \frac{\partial^2 f}{\partial x \partial y} \\ \frac{\partial^2 f}{\partial x \partial y} & \frac{\partial^2 f}{\partial^2 y} \end{bmatrix} \quad (1)$$

The two eigenvalues of H in equation (1) were then calculated. Negative and large eigenvalues suggest a bright and round local structure[32], i.e., a cell nucleus in our case. Hence the output image pixel value was set to either the product of the two eigenvalues if both were negative, or simply to zero otherwise. Last, a slight thresholding was applied to the output nuclear image to remove excessive background.

When a paraffin block is imaged, the paraffin also generates photoacoustic signals and appears as granular structures in the images. These spurious "nuclei" were extracted by the Hessian filter in the paraffin areas. In this way, we separated tissue from paraffin automatically in the images (Supplementary Fig. 4), based on the fact that tissue areas have a larger average pixel value and a smaller variation than paraffin areas. The local average and variation values were calculated for each pixel. The pixel was marked as tissue if the average was larger than an empirical threshold and the variation was smaller than an empirical threshold, or marked as paraffin otherwise. Then the spurious nuclei in the paraffin areas were removed.

The serial 2D images acquired by mPAM were converted to step-through videos by Amira or MATLAB for 3D visualization. Image co-registration was not needed because the mechanical scanning was stable and the specimen suffered no sectioning deformation while being imaged.

**Image analysis**. To evaluate the nuclear imaging results of mPAM, H&E images were used as the gold standard. We defined the nuclear sensitivity as the ratio of the number of true nuclei identified by mPAM to the number of all nuclei identified by H&E staining. Similarly, we defined the nuclear specificity as the ratio of the area of true non-nuclear tissue identified by mPAM to the area of all non-nuclear tissue identified by H&E, and defined nuclear PPV as the ratio of the number of nuclei that were true in the mPAM images to the number of all nuclei (including the spurious ones) identified in the mPAM images. To calculate the sensitivity, specificity, and PPV for Supplementary Fig. 2, we selected and analyzed four regions of the brain, each containing 100–200 nuclei.

To generate the nuclear density map, we first generated a nuclear image by Hessian filtering. Each nucleus in the image was reduced to one pixel with unit amplitude, and the background was set to zero amplitude. Then each pixel of this new image was replaced by the average of the $50 \times 50$ μm$^2$ surrounding area, creating a nuclear density map where each pixel value equaled the relative nuclear density of the $50 \times 50$ μm$^2$ surrounding area.

**Data availability**. The data that support the findings of this study are available from the corresponding author on reasonable request.

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

## Acknowledgements

The authors appreciate Prof. James Ballard's close reading of the manuscript and Yang Li's preparation of the system schematic. This work was sponsored in part by National Institutes of Health grants DP1 EB016986 (NIH Director's Pioneer Award) and R01 CA186567 (NIH Director's Transformative Research Award). Part of the work was performed at the Alafi Neuroimaging Laboratory, the Hope Center for Neurological Disorders, which is supported by the NIH Neuroscience Blueprint Center Core Grant P30 NS057105.

## Author contributions

T.T.W.W., R.Z., C.Z. and L.V.W. conceived of the study. T.T.W.W., R.Z., C.Z. and K.I.M. designed the imaging system. T.T.W.W., R.Z. and C.Z. built the imaging system. L.W. and J.S. wrote the operating software. R.C., K.K.S. and Q.Z. fabricated the ultrasonic transducer. T.T.W.W., R.Z. and C.Z. performed the imaging experiments. T.T.W.W. and R.Z. analyzed the data. T.T.W.W., R.Z., C.Z. and H.-C.H. processed the data. T.T.W.W. and L.V.W. wrote the manuscript. L.V.W. supervised the whole study.

## Additional information

**Competing interests:** L.V.W. has a financial interest in MicroPhotoAcoustics, CalPACT, LLC, and Union Photoacoustic Technologies, which, however, did not support this work. K.I.M. has a financial interest in MicroPhotoAcoustics. The remaining authors declare no competing financial interests.

