## [Peer Review file · Nature Communications]

Reviewers' comments:

Reviewer #1 (Remarks to the Author):

This paper describes a new application of photoacoustic microscopy to the important problem of the postmortem whole organ high resolution 3D imaging on a subcellular scale. Recently developed popular approaches to this problem combine automated microtome slicing of the organ, such as the brain, embedded in paraffin, resin, or a similar substance, followed by the optical microscopic imaging and computer based co-registering of multiple high resolution images. The authors propose to replace optical microscopy with label-free photoacoustic microscopy. This is certainly a very interesting and promising idea, as thin slicing often results in rips and tears, while staining can lead to stain inhomogeneity and crystallization. Photoacoustic microscopy can allow thicker and therefore less problematic slicing and also elimination of the stains. This is, therefore, very encouraging. At the same time, the authors need to clarify the following issue. Out of the two mentioned above problems of the existing approaches, the first one is considered much more difficult to overcome and, therefore, more important. It appears that photoacoustic microscopy is uniquely positioned to overcome this problem as it is much less affected by tissue scattering and, therefore, has a deeper penetration depth. However, instead of emphasizing this unique advantage of photoacoustic microscopy, the paper's main focus seems to be its label-free aspect, which is much less unique, as endogenous chromophores employed by photoacoustic microscopy are not selective. Therefore, I believe that refocusing the paper from its label-free aspect to the ability of photoacoustic microscopy to deal with thicker tissue sections would make it even stronger.

Reviewer #2 (Remarks to the Author):

This manuscript, "Label-free automated three-dimensional imaging of whole organs by microtomy-assisted photoacoustic microscopy", demonstrates the whole organ photoacoustic images with micro-scales. Developed microtomy-assisted PAM has great advantages in such large size imaging with distortion-free and registration-free using simple configurations. A similar approach has been already done using other microscopic modalities (e.g., Nat. Methods. 9, 255–258 (2012)) with tagging agents. However, the current manuscript is the first success in photoacoustic microscopy without using any agent. That means completely free from labelling. These two are the key novelties in this manuscript. The results are very remarkable and useful to applications in life sciences. Therefore, I strongly support the publication of this work on Nature Communications. To further improve the quality, I suggest the authors to address the following minor comments.

Comments

1. An imaging time of ~15 days (or 70 hours in 1/20 volume) is quite long. Does the laser take a rest during the imaging experiments? Is there any power drop of the laser to acquire one volumetric image?
2. How much energy of the pulse laser is used in PA imaging? Although this whole organ imaging is not in vivo experiment, this is somewhat important for readers to reproduce the experiments.
3. In Results section, 532 nm was used to show the lipid contrast but lipid absorption is not so high enough at 532 nm. The authors could select another wavelength using the OPO laser used in this experiment. Can the authors explain why?
4. In Methods section, sensitivity, specificity, and PPV were calculated for Supplementary Fig. 2 which is a sectioned tissue. Is it appropriate to calculate those parameters in the tissue block?
5. In Methods section, the authors claimed that "The imaging speed is limited by the laser repetition rate to 104 pixels per second, and an additional ~20 seconds are required for each

mechanical sectioning". But motorized stages should be also a limiting factor to slow down the imaging speed. Please clarify it for reader's understanding.

Reviewer #1

Summary: This paper describes a new application of photoacoustic microscopy to the important problem of the postmortem whole organ high resolution 3D imaging on a subcellular scale. Recently developed popular approaches to this problem combine automated microtome slicing of the organ, such as the brain, embedded in paraffin, resin, or a similar substance, followed by the optical microscopic imaging and computer based co-registering of multiple high resolution images. The authors propose to replace optical microscopy with label-free photoacoustic microscopy. This is certainly a very interesting and promising idea, as thin slicing often results in rips and tears, while staining can lead to stain inhomogeneity and crystallization. Photoacoustic microscopy can allow thicker and therefore less problematic slicing and also elimination of the stains. This is, therefore, very encouraging.

Response: We thank Reviewer #1 for the encouraging summary. We have addressed Reviewer #1's recommendation below.

Comment: At the same time, the authors need to clarify the following issue. Out of the two mentioned above problems of the existing approaches, the first one is considered much more difficult to overcome and, therefore, more important. It appears that photoacoustic microscopy is uniquely positioned to overcome this problem as it is much less affected by tissue scattering and, therefore, has a deeper penetration depth. However, instead of emphasizing this unique advantage of photoacoustic microscopy, the paper's main focus seems to be its label-free aspect, which is much less unique, as endogenous chromophores employed by photoacoustic microscopy are not selective. Therefore, I believe that refocusing the paper from its label-free aspect to the ability of photoacoustic microscopy to deal with thicker tissue sections would make it even stronger.

Response: We agree with Reviewer #1 that putting more emphasis on the advantage of photoacoustic microscopy for deep tissue imaging would make our paper even stronger. Therefore, we have added a new data (Supplementary Figure 8) showing that photoacoustic microscopy can provide deep penetration and depth-resolved information in a raster scan. This deep tissue imaging capability enables less slicing of the tissue, resulting in fewer sectioning artifacts. Correspondingly, we have made the following changes:

In the Abstract,

Added:

“Its deep tissue imaging capability leads to less sectioning, resulting in negligible sectioning artifact.”

In the Introduction, pg. 3,

Amended:

*“...100% endogenous natural staining in whole organs with **less sectioning and high fidelity.**”*

Added:

“With visible light illumination, mPAM shows its deep tissue imaging capability, which enables less slicing and hence reduces sectioning artifacts.”

In the Results section, under sub-title Imaging formalin-fixed agarose-embedded mouse brains, pg. 4,

Added:

*“To show that deep tissue imaging can also be achieved with mPAM, we used 420 nm light to illuminate a 1 mm thick mouse brain slice. A representative xz projected image of the mouse brain is provided in **Supplementary Fig. 8**, which shows cytochrome contrast based structures ~800 μm deep. Therefore, where cytochrome is the target of interest, we can further reduce the number of slices, resulting in even fewer sectioning artifacts.”*

Reviewer #2

Summary: This manuscript, “Label-free automated three-dimensional imaging of whole organs by microtomy-assisted photoacoustic microscopy”, demonstrates the whole organ photoacoustic images with micro-scales. Developed microtomy-assisted PAM has great advantages in such large size imaging with distortion-free and registration-free using simple configurations. A similar approach has been already done using other microscopic modalities (e.g., Nat. Methods. 9, 255–258 (2012)) with tagging agents. However, the current manuscript is the first success in photoacoustic microscopy without using any agent.

That means completely free from labelling. These two are the key novelties in this manuscript. The results are very remarkable and useful to applications in life sciences. Therefore, I strongly support the publication of this work on Nature Communications. To further improve the quality, I suggest the authors to address the following minor comments.

Response: We thank Reviewer #2 for the supportive summary. We have addressed Reviewer #2's recommendations one-by-one as follows.

Comment 1: An imaging time of ~15 days (or 70 hours in 1/20 volume) is quite long. Does the laser take a rest during the imaging experiments? Is there any power drop of the laser to acquire one volumetric image?

Response 1: We appreciate the question. We took this laser overloading issue into consideration when we performed the imaging experiments. To reduce the burden on the laser, there was always a 30-minute break between consecutive raster scans. With this precaution, we did not observe any drop of the laser power after all the imaging experiments. To clarify this point, we have added the following text to the Methods section, under the sub-title mPAM system,

“Note that, to avoid laser overload, we paused the laser for 30 minutes between consecutive raster scans. We did not observe any power drop of the laser throughout acquisition of all the volumetric images.”

Comment 2: How much energy of the pulse laser is used in PA imaging? Although this whole organ imaging is not in vivo experiment, this is somewhat important for readers to reproduce the experiments.

Response 2: We agree that we should clarify how much energy of the pulse laser is used in PA imaging, even if it was not an *in vivo* experiment. We have added the following text to the Methods section, under the sub-title mPAM system,

“...with a 266 nm wavelength, 7 ns pulse width, ~5 nJ pulse energy on the imaging targets, and 10 kHz pulse repetition rate...”

And,

“...with a 420 nm wavelength, 5 ns pulse width, ~200 nJ pulse energy on the imaging targets, and 1 kHz pulse repetition rate...”

Comment 3: In Results section, 532 nm was used to show the lipid contrast but lipid absorption is not so high enough at 532 nm. The authors could select another wavelength using the OPO laser used in this experiment. Can the authors explain why?

Response 3: We thank you for the apt suggestion. We agree that, if possible, we should use a more suitable laser wavelength to probe the targeted biomolecules. The primarily reason is that our OPO laser is not providing enough laser energy at the optimal wavelength. To show a similar claim, we have re-done the experiment with 420 nm (instead of 532 nm) to image cytochrome at its optimal absorption wavelength. We have accordingly modified Supplementary Figure 7, and changed the text in different paragraphs to address the change of wavelength (from 532 nm to 420 nm):

All “532 nm” are now changed to “420 nm”.

Also, in the Result section, under the sub-title Imaging formalin-fixed agarose-embedded mouse brains, pg. 4,

“To show that mPAM can image more endogenous biomolecules, we used dual-wavelength illumination (266 and 420 nm) to image an agarose-embedded brain slice (Supplementary Fig. 7(a), (b)). With 266 nm laser illumination, Supplementary Fig. 7(a) shows mostly DNA/RNA and lipid contrasts, whereas with 420 nm laser illumination, Supplementary Fig. 7(b) shows mostly cytochrome contrast. The overlay image (Supplementary Fig. 7(c)) is displayed in two-channel pseudo colors, which represent the optical absorption color contrasts of the biomolecules at the two wavelengths and illustrates that more biomolecules are imaged by dual-wavelength mPAM than by single-wavelength mPAM.”

And Supplementary Figure 7’s caption,

“...(b) Label-free mPAM image with 420 nm laser illumination, which shows cytochrome (C) contrast...”

Comment 4: In Methods section, sensitivity, specificity, and PPV were calculated for Supplementary Fig. 2 which is a sectioned tissue. Is it appropriate to calculate those parameters in the tissue block?

Response 4: We thank you for raising this question. The reason why we could calculate these parameters (sensitivity, specificity, and PPV) in a sectioned tissue is because we could have the corresponding gold standard H&E stained image to compare with. Therefore, it is highly appropriate to calculate those parameters in a thin sectioned tissue, confirming that information can be extracted from our mPAM images with high accuracy. However, in a tissue block, it is not feasible to get a thick tissue H&E stained image. Therefore, we do not have a ground truth for comparison. We believe it would not be appropriate

to calculate those parameters in a tissue block. Nevertheless, given the accurate result in a thin section and the fact that no additional sample perturbation is needed for mPAM, we expect the actual sensitivity, specificity, and PPV are also high in a tissue block. This high accuracy claim can be proven by Supplementary Figure 5, which shows that the nuclear distributions in the mPAM and H&E images are strongly correlated in a tissue block setting.

Comment 5: In Methods section, the authors claimed that “The imaging speed is limited by the laser repetition rate to 10^4 pixels per second, and an additional ~20 seconds are required for each mechanical sectioning”. But motorized stages should be also a limiting factor to slow down the imaging speed. Please clarify it for reader’s understanding.

Response 5: We agree that we should clarify whether the motorized stages are a limiting factor in the imaging speed. According to the specification of the motorized stages that we used, the maximum velocity is ~50 mm/s (<https://www.physikinstrumente.com/en/products/linear-stages-and-actuators/stages-with-motor-screw-drives/pls-85-precision-linear-stage-601301/>). For our current design, the lateral scanning step size is 0.625 μm . With 10^4 pixels per second, the required velocity is ~6.25 mm/s, which is much less than the maximum velocity of the motorized stages (~50 mm/s). Therefore, for our case, the motorized stages are not the limiting factor. To clarify this for the reader’s understanding, we amended the text in the Methods section and put the amended text under the sub-title mPAM system,

“With our current design and setting, the imaging speed is limited by the laser repetition rate to 10^4 pixels per second. The lateral scanning step size is 0.625 μm . At 10^4 pixels per second, the required scanning speed of a motorized stage is ~6.25 mm/s. The motorized stage that we used for the fast scanning axis (PLS-85) can scan at up to 50 mm/s. Therefore, the motorized stage is not the limiting factor on the imaging speed.”

REVIEWERS' COMMENTS:**Reviewer #1 (Remarks to the Author):**

The authors have revised the manuscript, emphasizing the ability of photoacoustic microscopy to deal with thicker tissue sections. They also added new data illustrating the above concept. I feel that my main concern was adequately addressed and, therefore, recommend publication.

Reviewer #2 (Remarks to the Author):

The authors have very well addressed my previous minor comments. This manuscript is ready to be published in Nat. Comm. and will attract great attention from a broad range of readers.

Reviewer #1

Comment: The authors have revised the manuscript, emphasizing the ability of photoacoustic microscopy to deal with thicker tissue sections. They also added new data illustrating the above concept. I feel that my main concern was adequately addressed and, therefore, recommend publication.

Reviewer #2

Comment: The authors have very well addressed my previous minor comments. This manuscript is ready to be published in Nat. Comm. and will attract great attention from a broad range of readers.

Response: We thank Reviewers #1 and #2 for their supportive comments.